# Efficient Estimation of Local Robustness of Machine Learning Models

**Tessa Han** [1]   **Suraj Srinivas** [1]   **Himabindu Lakkaraju** [1]

## Abstract

Machine learning models often need to be robust to noisy input data. The effect of real-world noise (which is often random) on model predictions is captured by a model's local robustness, i.e., the consistency of model predictions in a local region around an input. Local robustness is therefore an important characterization of real-world model behavior and can be useful for debugging models and establishing user trust. However, the naïve approach to computing local robustness based on Monte-Carlo sampling is statistically inefficient, leading to prohibitive computational costs for large-scale applications. In this work, we develop the first analytical estimators to efficiently compute local robustness of multi-class discriminative models using local linear function approximation and the multivariate Normal CDF. Through the derivation of these estimators, we show how local robustness is connected to concepts such as randomized smoothing and softmax probability. We also confirm empirically that these estimators accurately and efficiently compute the local robustness of standard deep learning models. In addition, we demonstrate these estimators' usefulness for various tasks involving local robustness, such as measuring robustness bias and identifying examples that are vulnerable to noise perturbation in a dataset. y developing analytical estimators of local robustness, but also makes its computation practical, enabling the use of local robustness in critical downstream applications.

## 1. Introduction

A desirable attribute of machine learning models is robustness to perturbations of input data. One common notion of robustness is adversarial robustness, the ability of a model to maintain its prediction when presented with adversarial perturbations, i.e., perturbations designed to cause the model to change its prediction. Although adversarial robustness can identify whether an adversarial example exists in a local region around an input, real-world noise (such as measurement noise) is rarely adversarial and often random. The effect of such noise on model predictions is better captured by another notion of robustness: *local robustness*, the fraction of points in a local region around an input for which the model provides consistent predictions. This is a strict generalization of adversarial robustness – if the fraction of points for which the model is consistent is less than 1, then an adversarial perturbation exists. The limitation of adversarial robustness in only detecting whether an adversarial perturbation exists is perhaps expected, as its original use-case was motivated by model security, not model understanding, debugging, or regularization for improved generalization. Therefore, local robustness provides a more comprehensive characterization of real-world model behavior as it captures model behavior under the interference of average case noise.

In this paper, we take the first steps towards measuring local robustness. We show that the naïve approach to estimating local robustness based on Monte-Carlo sampling (Nanda et al., 2021) is statistically inefficient and impractical: obtaining an accurate estimate of local robustness using this approach requires a large number of samples from the local region. This inefficiency, which is exacerbated in the case of high-dimensional data, leads to prohibitive computational costs for large-scale applications.

To address this problem, we develop the first analytical estimators to efficiently compute the local robustness of a model. More specifically, our work makes the following contributions:

1. We derive a series of novel analytical estimators to efficiently compute the local robustness of multi-class discriminative models by linearizing non-linear models in the local region around an input, and computing the model's local robustness in this region using the multivariate Normal cumulative distribution function. Through the derivation, we show how local robustness is connected to such concepts as randomized smoothing and softmax probability.

---

[1]Harvard University, Cambridge, MA. Correspondence to: Tessa Han <than@g.harvard.edu>, Suraj Srinivas <ssrinivas@seas.harvard.edu>, Himabindu Lakkaraju <hlakkaraju@hbs.edu>.

*Workshop on Interpretable ML in Healthcare at International Conference on Machine Learning (ICML)*, Honolulu, Hawaii, USA. 2023. Copyright 2023 by the author(s).

2. We empirically validate these analytical estimators using standard deep learning models and datasets, demonstrating that these estimators accurately and efficiently compute local robustness.

3. We demonstrate the usefulness of our analytical estimators for various tasks involving local robustness, such as measuring class-level robustness bias (Nanda et al., 2021) (i.e., a model being more locally robust for some classes than for others) and identifying examples that are vulnerable to noise perturbation in a dataset. Such dataset-level analyses of local robustness are made practical only by having these efficient analytical estimators.

To our knowledge, this work is the first to investigate local robustness in a multi-class setting and develop efficient analytical estimators for local robustness. The analytical aspect of these estimators not only advances conceptual understanding of local robustness, but also enables local robustness to be used in applications that require differentiability (such as model training). In addition, the efficiency of these estimators makes the computation of local robustness practical, enabling tasks that assist in such important objectives as debugging models and establishing user trust.

## 2. Related Work

**Linearization of neural networks.** Prior works have used linear models to approximate neural networks in the local region around an input. For example, LIME (Ribeiro et al., 2016), a popular post hoc explanation method, does so and uses the coefficients of the linear model as feature attributions. In addition, the local function approximation framework (Han et al., 2022) demonstrates that eight popular post hoc explanation methods all perform local linear approximation of the underlying model. Local linear function approximation has also been used to generate probabilistically-robust counterfactual explanations, specifying the probability that a binary classifier generates consistent predictions when inputs are noisy (Pawelczyk et al., 2023). In contrast to these prior works which apply local linear function approximation to post hoc explainability (Ribeiro et al., 2016; Han et al., 2022; Pawelczyk et al., 2023) or focus on binary classification (Pawelczyk et al., 2023), this work uses local linear function approximation to investigate local robustness and develops analytical estimators of local robustness for both binary and multi-class classification.

**Adversarial robustness.** Prior works have proposed methods to generate adversarial attacks (Carlini and Wagner, 2017; Goodfellow et al., 2015; Moosavi-Dezfooli et al., 2016), which detect whether an adversarial perturbation exists in a local region, and methods to certify model robustness (Cohen et al., 2019; Carlini et al., 2022), which provide guarantees of dataset-level robustness (i.e., all points in a dataset are robust to a certain amount of noise). In contrast to these prior works on adversarial robustness, this work investigates local robustness (a generalization of adversarial robustness), which calculates the proportion of points in a local region for which model predictions are consistent and measures point-level robustness. Prior work has also investigated robustness bias in terms of vulnerability to adversarial attacks, lowerbounding and upperbounding a point's probability of adversarial attack (Nanda et al., 2021) using sampling approaches. In contrast, this work investigates robustness bias in terms of local robustness, directly calculating a model's local robustness using analytical approaches.

**Uncertainty estimation.** Prior works have also developed approaches to measure a model's prediction uncertainty. These include calibration (Guo et al., 2017), Bayesian uncertainty (Kendall and Gal, 2017), and conformal prediction (Shafer and Vovk, 2008). In contrast to these prior works in which prediction uncertainty is with respect to a calibration set (Guo et al., 2017; Shafer and Vovk, 2008) or model parameters (Kendall and Gal, 2017), this work investigates local robustness, which can be thought of as prediction uncertainty with respect to input noise.

## 3. Our Framework: The Local Robustness Estimator Family

In this section, we first describe the mathematical problem of local robustness estimation. Then, we present the naïve estimator based on sampling and derive more efficient analytical estimators. Lastly, we explore the connections between local robustness and softmax probability.

### 3.1. Notation and Preliminaries

Assume we have a neural network $f : \mathbb{R}^d \rightarrow \mathbb{R}^C$ with $C$ output classes, and the model predicts class $t \in [1, ...C]$ for a given input $\mathbf{x} \in \mathbb{R}^d$, i.e., $t = \arg\max_{i=1}^{C} f_i(\mathbf{x})$, where $f_i$ denotes the logits for the $i^{th}$ class. Given this model, the local robustness estimation problem is to compute the probability of consistent classification (to class $t$) under noise perturbation of the inputs.

*Definition* 1. We define the average local robustness of a classifier $f$ at a point $\mathbf{x}$ as the probability of being classified to class $t$ under Normal noise $\mathcal{N}(0, \sigma^2)$ added to the inputs, denoted as

$$p_\sigma^{\text{robust}}(\mathbf{x}, t) = P_{\epsilon \sim \mathcal{N}(0,\sigma^2)}\left[\arg\max_i f_i(x + \epsilon) = t\right]$$

Thus, the higher $p_\sigma^{\text{robust}}(\mathbf{x}, t)$ is, the more robust the model is in the local neighborhood around $\mathbf{x}$. In this paper, given that we always measure local robustness with respect to the

predicted class $t$ at $\mathbf{x}$, we henceforth suppress the dependence on $t$ in the notation.

Note that $p_\sigma^{\text{robust}}$ *generalizes adversarial robustness*. Adversarial robustness detects the presence absence of a perturbation in a local neighborhood that leads to misclassification, while local robustness computes the probability of consistent classification. In other words, adversarial robustness is concerned with the quantity $\mathbf{1}(p_\sigma^{\text{robust}} < 1)$, i.e., the indicator function that local robustness is less than one (which indicates the presence of an adversarial perturbation), while local robustness is concerned with the quantity $p_\sigma^{\text{robust}}$ itself. In the rest of this section, we derive estimators for $p_\sigma^{\text{robust}}$.

### 3.1.1. ESTIMATOR 0: THE MONTE-CARLO ESTIMATOR $p_\sigma^{\text{mc}}$

The most naïve estimator of average local robustness is a Monte-Carlo estimator, i.e.,

$$
\begin{aligned}
p_\sigma^{\text{robust}}(\mathbf{x}) &= P_{\epsilon \sim \mathcal{N}(0,\sigma^2)} \left[ \arg\max_i f_i(x+\epsilon) = t \right] \\
&= \mathbb{E}_{\epsilon \sim \mathcal{N}(0,\sigma^2)} \left[ \mathbf{1}_{\arg\max_i f_i(x+\epsilon)=t} \right] \\
&\approx \frac{1}{M} \sum_{j=1}^{M} \left[ \mathbf{1}_{\arg\max_i f_i(x+\epsilon_j)=t} \right] = p_\sigma^{\text{mc}}(\mathbf{x})
\end{aligned}
$$

$p_\sigma^{\text{mc}}$ replaces the expectation with the sample average using Monte-Carlo sampling and has been used in prior work (Nanda et al., 2021). While Monte-Carlo estimators are technically independent of dimensionality (Vershynin, 2018), in practice, for typical use cases involving neural networks, this estimator requires a large number of random samples to converge to the underlying expectation. For example, for MNIST and CIFAR10 CNNs, it takes around $M = 10,000$ samples per point for $p_\sigma^{\text{mc}}$ to converge, which is computationally infeasible. Thus, we set out to address this problem by developing more efficient estimators.

## 3.2. Analytical Estimators of Local Robustness

### 3.2.1. ESTIMATOR 1: THE TAYLOR ESTIMATOR $p_\sigma^{\text{taylor}}$

Our goal is to derive analytical estimators which reduce the complexity of estimating local robustness. To this end, we first locally linearize non-linear models and then compute the local robustness of the resulting linear models. However, even the problem of computing the local robustness of linear models is more challenging than it appears due to the complex geometry of linear decision boundaries given $C$ classes. In particular, the relative orientation and similarities of these class-wise decision boundaries needs to be taken into account to compute local robustness.

Specifically, given a linear model for a three-class classifica-

tion problem with weights $w_1, w_2, w_3$ and biases $b_1, b_2, b_3$, such that $y = \arg\max_i \{ w_i^\top \mathbf{x} + b_i \mid i \in [1,2,3] \}$, the decision boundary between classes 1 and 2 is given by $y_{12} = (w_1 - w_2)^\top \mathbf{x} + (b_1 - b_2)$. This is easy to verify as for any $\mathbf{x}$ such that $y_{12} = 0$, we have $w_1^\top \mathbf{x} + b_1 = w_2^\top \mathbf{x} + b_2$, making it the decision boundary. Thus, the relevant quantities are the pairwise difference terms among the weights and biases which characterize the decision boundaries. We take this into account and provide the expression for the linear case below.

**Lemma 3.1.** *The local robustness of a multi-class linear model $f(\mathbf{x}) = \mathbf{w}^\top \mathbf{x} + b$, with $\mathbf{w} \in \mathbb{R}^{d \times C}$ and $b \in \mathbb{R}^C$, with respect to a target class $t$ is given by the following. Define the decision boundary weights $w_i' = w_t - w_i \in \mathbb{R}^d, \forall i \neq t$, where $w_t, w_i$ are rows of $\mathbf{w}$ and biases $b_i' = (w_t' - w_i')^\top \mathbf{x} + (b_t - b_i) \in \mathbb{R}$, then*

$$
p_\sigma^{\text{robust}}(\mathbf{x}) = CDF_{\mathcal{N}(0,UU^\top)} \left( \frac{b_1'}{\sigma \|w_1'\|_2}, \dots \frac{b_i'}{\sigma \|w_i'\|_2}, \dots \frac{b_C'}{\sigma \|w_C'\|_2} \right)
$$

$$
\text{where} \quad U = \left[ \frac{w_1'}{\|w_1'\|_2}; \dots \frac{w_i'}{\|w_i'\|_2}; \dots \frac{w_C'}{\|w_C'\|_2} \right] \in \mathbb{R}^{(C-1) \times d}
$$

*and $CDF_{\mathcal{N}(0,UU^\top)}$ refers to the $(C-1)$-dimensional Normal CDF with covariance $UU^\top$.*

The proof is in Appendix A.1. The matrix $U$ exactly captures the geometry of the linear decision boundaries and the covariance matrix $UU^\top$ encodes the relative similarity between pairs of decision boundaries. If the decision boundaries are all orthogonal to each other, then the covariance matrix is the identity matrix. However, we find that, in practice, the covariance matrix is strongly non-diagonal, indicating that the decision boundaries are not orthogonal to each other.

For diagonal covariance matrices, the multivariate Normal CDF (*mvn-cdf*) can be written as the product of univariate Normal CDFs, which would be easy to compute. However, the strong non-diagonal nature of covariance matrices in practice leads to the resulting *mvn-cdf* not having a simple closed form solution, with the only alternative being approximation of the integral via sampling (Botev, 2017; Sci). However, this sampling is performed in the $(C-1)$-dimensional space as opposed to the $d$-dimensional space that $p_\sigma^{\text{mc}}$ performs. In practice, for classification problems, we often have $C << d$, making sampling in $(C-1)$-dimensions more efficient. Using the derived robustness estimator for the linear model, we now derive the Taylor estimator.

*Proposition* 1. The Taylor estimator for a model $f$ and point $\mathbf{x}$ is given by linearizing $f$ around $\mathbf{x}$ using $\mathbf{w} = \nabla_{\mathbf{x}} f(\mathbf{x})$ and $b = f(\mathbf{x})$, with decision boundaries $g_i(\mathbf{x}) = f_t(\mathbf{x}) - f_i(\mathbf{x})$, $\forall i \neq t$, leading to

$$
p_\sigma^{\text{taylor}}(\mathbf{x}) = \text{CDF}_{\mathcal{N}(0,UU^\top)} \left( \left[ \frac{g_1(\mathbf{x})}{\sigma \|\nabla_{\mathbf{x}} g_1(\mathbf{x})\|_2}, \dots, \frac{g_C(\mathbf{x})}{\sigma \|\nabla_{\mathbf{x}} g_C(\mathbf{x})\|_2} \right] \right)
$$

with $U \in \mathbb{R}^{(C-1) \times d}$ defined as in the linear case.

The proof is in Appendix A.1. The smaller the local region around $\mathbf{x}$, the more faithful the local linearization of the model. Thus, the smaller the $\sigma$, the more accurate the Taylor estimator is.

### 3.2.2. ESTIMATOR 2: THE MMSE ESTIMATOR $p_\sigma^{\mathbf{mmse}}$

While the Taylor estimator is more efficient than the naïve one, it has a drawback. In particular, its linear approximation is valid near the data point, but gets worse farther away from the data point. To fix this issue, we use a linearization that is faithful to the model on the entire noise distribution, not just near the data point. Linearization has been studied in feature attribution research, which concerns itself with approximating non-linear models with linear ones to produce model explanations (Han et al., 2022). In particular, the SmoothGrad (Smilkov et al., 2017) technique has been described as the MMSE optimal linearization of the model (Han et al., 2022; Agarwal et al., 2021). Using similar techniques, we propose the MMSE estimator $p_\sigma^{\mathrm{mmse}}$ as follows.

*Proposition* 2. The MMSE estimator for a model $f$ and point $\mathbf{x}$ is given by linearizing $f$ around $\mathbf{x}$ using $\mathbf{w} = \sum_{j=1}^{N} \nabla_{\mathbf{x}} f(\mathbf{x} + \epsilon)$ and $b = \sum_{j=1}^{N} f(\mathbf{x})$, with decision boundaries $g_i(\mathbf{x}) = f_t(\mathbf{x}) - f_i(\mathbf{x}), \forall i \neq t$, leading to

$$p_\sigma^{\mathrm{mmse}}(\mathbf{x}) = \mathrm{CDF}_{\mathcal{N}(0, UU^\top)} \left( \left[ \frac{\frac{1}{N} \sum_{j=1}^{N} g_1(\mathbf{x}+\epsilon)}{\sigma \| \frac{1}{N} \sum_{j=1}^{N} \nabla_{\mathbf{x}} g_1(\mathbf{x}+\epsilon) \|_2}, \right. \right.$$
$$\left. \left. \dots \frac{\frac{1}{N} \sum_{j=1}^{N} g_C(\mathbf{x}+\epsilon)}{\sigma \| \frac{1}{N} \sum_{j=1}^{N} \nabla_{\mathbf{x}} g_C(\mathbf{x}+\epsilon) \|_2} \right] \right)$$

with $U \in \mathbb{R}^{(C-1) \times d}$ defined as in the linear case, where $N$ is the number of perturbations.

The proof is in Appendix A.1. It involves creating a randomized smooth model (Cohen et al., 2019) from the base model and then computing the decision boundaries of this smooth model. Note that the gradients of this smooth model are equal to those obtained from SmoothGrad. We show, for the first time, that performing such randomization helps compute robustness information for the original base model.

Like $p_\sigma^{\mathrm{mc}}$, $p_\sigma^{\mathrm{mmse}}$ also requires sampling over the input space like $p_\sigma^{\mathrm{mc}}$. However, due to its reliance on model gradients, it requires far fewer number of samples to converge (we observed around $N = 5 - 10$ to suffice in practice), thus making it computationally efficient.

### 3.2.3. ESTIMATORS 3 & 4 : APPROXIMATE TAYLOR AND MMSE ESTIMATORS

One drawback of the Taylor and MMSE estimators is their use of the *mvn-cdf*, which does not have a closed form solution and can cause the estimators to be slow for problems with a large number of classes $C$. Further, the *mvn-cdf* makes these estimators non-differentiable, which is inconvenient for applications which require differentiating $p_\sigma^{\mathrm{robust}}$. To alleviate these issues, we wish to approximate the *mvn-cdf* with an analytical closed-form expression. As CDFs are monotonically increasing functions, the approximation should also be monotonically increasing.

To this end, we find that the *univariate* Normal CDF is well-approximated by the sigmoid function, and has been used to propose the GeLU activation function (Hendrycks and Gimpel, 2016). Inspired by this, we propose to approximate the *mvn-cdf* with the multivariate-sigmoid function, which we define as follows:

*Definition* 2. The multivariate sigmoid is defined as $\mathrm{mv\text{-}sigmoid}(\mathbf{x}) = \frac{1}{1 + \sum_i \exp(-\mathbf{x}_i)}$

We find experimentally that *mv-sigmoid* approximates the *mvn-cdf* well for practical values of the covariance matrix $UU^\top$. Using this approximation to substitute *mv-sigmoid* for the *mvn-cdf* in the $p_\sigma^{\mathrm{taylor}}$ and $p_\sigma^{\mathrm{mmse}}$ estimators, we get estimators $p_\sigma^{\mathrm{taylor\_mvs}}$ and $p_\sigma^{\mathrm{mmse\_mvs}}$, respectively.

### 3.3. Exploring the Connections Between Local Robustness and Softmax Probability

#### 3.3.1. ESTIMATOR 5: SOFTMAX AS AN ESTIMATOR OF $p_\sigma^{\mathbf{robust}}$

Lastly, we observe that for linear models with a specific noise perturbation $\sigma$, the common softmax function taken with respect to the output logits can be viewed as an estimator of $p_\sigma^{\mathrm{robust}}$, albeit in a very restricted setting. Specifically,

**Lemma 3.2.** *For linear models* $f(\mathbf{x}) = \mathbf{w}^\top \mathbf{x} + b$, *such that the decision boundary weight norms* $\|w_i'\|_2 = \|w_j'\|_2 = \|w\|_2, \forall i, j$, *we have*

$$p_T^{softmax} = p_\sigma^{taylor\_mvs} \quad where \quad T = \sigma \|w\|_2$$

*Proof.* Let us consider softmax with respect to the $t^{th}$ output class and define $g_i(\mathbf{x}) = f_t(\mathbf{x}) - f_i(\mathbf{x})$, with $f$ being the linear model logits. Using this, we first show that softmax is identical to *mv-sigmoid*:

$$p_T^{\mathrm{softmax}}(\mathbf{x}) = \mathrm{softmax}_t(f_1(\mathbf{x})/T, ..., f_C(\mathbf{x})/T)$$
$$= \frac{\exp(f_t(\mathbf{x})/T)}{\sum_i \exp(f_i(\mathbf{x})/T)}$$
$$= \frac{1}{1 + \sum_{i; i \neq t} \exp((f_i(\mathbf{x}) - f_t(\mathbf{x}))/T)}$$
$$= \mathrm{mv\text{-}sigmoid} [g_1(\mathbf{x})/T, g_2(\mathbf{x})/T, ...g_C(\mathbf{x})/T]$$

Next, by denoting $w_i' = w_t - w_i$, each row has equal norm $\|w_i'\|_2 = \|w_j'\|_2, \forall i, j, t \in [1, ...C]$ which implies:

$$p_\sigma^{\text{taylor\_mvs}}(\mathbf{x})$$

$$= \text{mv-sigmoid}\left[\frac{g_1(\mathbf{x})}{\sigma\|w_1'\|_2}, \dots \frac{g_C(\mathbf{x})}{\sigma\|w_C'\|_2}\right]$$

$$= \text{mv-sigmoid}\left[g_1(\mathbf{x})/T, \dots, g_C(\mathbf{x})/T\right] \quad \because T = \sigma\|w_i'\|_2$$

$$= p_T^{\text{softmax}}(\mathbf{x})$$

□

This indicates that the temperature parameter $T$ of softmax roughly corresponds to the $\sigma$ of the added Normal noise with respect to which local robustness is measured. Overall, this shows that under the restricted setting where the local linear model consists of decision boundaries with equal weight norms, the softmax outputs can be viewed as an estimator of the $p_\sigma^{\text{taylor\_mvs}}$ estimator, which itself is an estimator of $p_\sigma^{\text{robust}}$. However, due to the multiple levels of approximation, we can expect the quality of $p_T^{\text{softmax}}$'s approximation of $p_\sigma^{\text{robust}}$ to be poor in general settings (outside of the very restricted setting), so much so that in general settings, $p_\sigma^{\text{robust}}$ and $p_T^{\text{softmax}}$ would be unrelated.

## 4. Experimental Evaluation

In this section, we describe our empirical evaluation. First, we evaluate the accuracy and efficiency of the analytical estimators. Then, we analyze the relationship between local robustness and softmax probability. Lastly, we demonstrate the usefulness of the estimators in real-world applications.

**Datasets and Models.** We evaluate the estimators on four datasets: MNIST (Deng, 2012), FashionMNIST (Xiao et al., 2017), CIFAR10 (Krizhevsky et al., 2009), and CIFAR100 (Krizhevsky et al., 2009). For MNIST and FashionMNIST, we train a linear model and a CNN to perform 10-class classification. For CIFAR10 and CIFAR100, we train a ResNet18 (He et al., 2016) model to perform 10-class and 100-class classification, respectively. We train the ResNet18 models using varying levels of gradient norm regularization ($\lambda$) to obtain models with varying levels of robustness. For the experiments below, we use 1,000 randomly-selected points from each dataset's full 10,000-point test set. Additional details about the datasets and models are described in Appendix A.2 and A.3.

### 4.1. Evaluation of the accuracy of analytical estimators

**The analytical estimators accurately compute local robustness.** To confirm that the analytical estimators accurately compute $p_\sigma^{\text{robust}}$, we calculate $p_\sigma^{\text{robust}}$ for each model and test set using $p_\sigma^{\text{mc}}$, $p_\sigma^{\text{taylor}}$, $p_\sigma^{\text{mmse}}$, $p_\sigma^{\text{taylor\_mvs}}$, $p_\sigma^{\text{mmse\_mvs}}$, and $p_T^{\text{softmax}}$ for different $\sigma$'s. For $p_\sigma^{\text{mc}}$, $p_\sigma^{\text{mmse}}$, and $p_\sigma^{\text{mmse\_mvs}}$, we use a sample size at which these estima-

tors have converged ($n = 10000, 1000,$ and $1000$). Convergence analyses are described in Appendix A.4 (henceforth, these estimators use these sample sizes). Then, we measure the absolute and relative difference between $p_\sigma^{\text{mc}}$ and the other estimators. The smaller these differences, the more accurately the estimator computes $p_\sigma^{\text{robust}}$. The performance of estimators for the FashionMNIST CNN model is shown in Figure 1.

The results indicate that $p_\sigma^{\text{mmse\_mvs}}$ and $p_\sigma^{\text{mmse}}$ are the best estimators of $p_\sigma^{\text{robust}}$, followed closely by $p_\sigma^{\text{taylor\_mvs}}$ and $p_\sigma^{\text{taylor}}$, and trailed by $p_T^{\text{softmax}}$. Consistent with the theory in Section 3, the MMSE estimators outperform the Taylor ones because the former obtains better estimates of $\nabla_{\mathbf{x}} g_i(\mathbf{x})$, and $p_T^{\text{softmax}}$ performs poorly in general settings because of its multiple levels of approximation.

The results also show that the smaller the noise neighborhood $\sigma$, the more accurately the estimators compute $p_\sigma^{\text{robust}}$. For the MMSE and Taylor estimators, this is because their linear approximation of the model around the input is more accurate for smaller $\sigma$'s. As expected, when the model is linear, $p_\sigma^{\text{taylor}}$ and $p_\sigma^{\text{mmse}}$ accurately compute $p_\sigma^{\text{robust}}$ for all $\sigma$'s (Appendix A.4). For $p_T^{\text{softmax}}$, values are constant across $\sigma$ and this particular model has high $p_T^{\text{softmax}}$ values for most points. Thus, for small $\sigma$'s where $p_\sigma^{\text{robust}}$ is near one, $p_T^{\text{softmax}}$ happens to approximate $p_\sigma^{\text{robust}}$ for this particular model.

**For robust models, the analytical estimators compute local robustness more accurately over a larger noise neighborhood.** The performance of $p_\sigma^{\text{mmse}}$ for CIFAR10 ResNet18 models of varying levels of robustness is shown in Figure 2. The results indicate that for more robust models (larger $\lambda$), the estimator is more accurate over a larger $\sigma$. This is because gradient norm regularization leads to models that are more locally linear, making the estimator's linear approximation of the model around the input more accurate over a larger $\sigma$.

**The mv-sigmoid function approximates the multivariate Normal CDF well in practice.** To examine *mv-sigmoid*'s approximation of *mvn-cdf*, we compute both functions using the same inputs ($z = \left[\frac{g_1(\mathbf{x})}{\sigma\|\nabla_{\mathbf{x}} g_1(\mathbf{x})\|_2}, \dots, \frac{g_C(\mathbf{x})}{\sigma\|\nabla_{\mathbf{x}} g_C(\mathbf{x})\|_2}\right]$, as described in Proposition 1) for the CIFAR10 ResNet18 model and its test set for different $\sigma$'s. The plot of *mv-sigmoid(z)* against *mvn-cdf(z)* for $\sigma = 0.1$ is shown in Figure 3. The results indicate that the two functions are strongly positively correlated, suggesting that *mv-sigmoid* approximates the *mvn-cdf* well in practice.

### 4.2. Evaluation of the efficiency of analytical estimators

**The naïve estimator is statistically inefficient.** To examine the efficiency of $p_\sigma^{\text{mc}}$, we calculate $p_\sigma^{\text{mc}}$ for each model and test set using different sample sizes ($n$) over different $\sigma$'s,

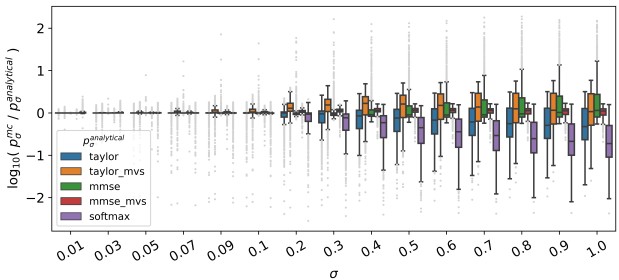

*Figure 1.* Experimental validation of analytical estimators (FashionMNIST CNN). The smaller the $\sigma$, the more accurately the estimators compute $p_\sigma^{\text{robust}}$. $p_\sigma^{\text{mmse}}$ and $p_\sigma^{\text{mmse\_mvs}}$ are the best estimators of $p_\sigma^{\text{robust}}$.

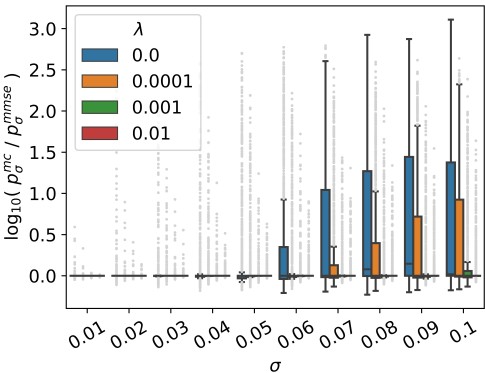

*Figure 2.* Experimental validation of analytical estimators (CIFAR10 ResNet18). For more robust models, the estimators compute $p_\sigma^{\text{robust}}$ more accurately over a larger noise neighborhood.

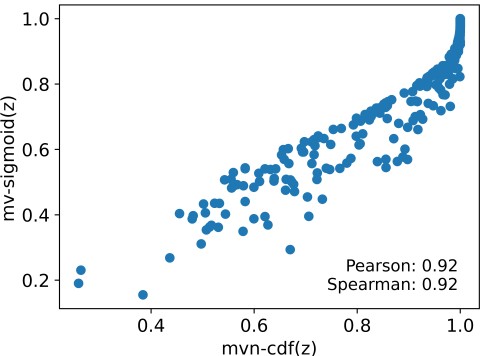

*Figure 3.* Correlation of *mvn-cdf(z)* and *mv-sigmoid(z)* for the CIFAR10 ResNet18 model. The formulation of $z$ is described in Section 4.1. In practice, *mv-sigmoid* approximates *mvn-cdf* well.

and measure the absolute and relative difference between

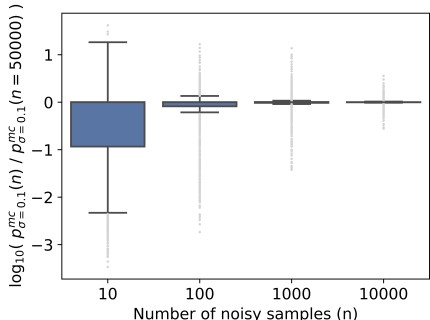

*Figure 4.* Convergence of the naïve estimator $p_\sigma^{\text{mc}}$ for the CIFAR10 ResNet18 model as the number of noisy samples increases. In practice, $p_\sigma^{\text{mc}}$ is statistically inefficient.

| Estimator | CPU: Intel x86_64 | | GPU: Tesla V100 | |
|---|---|---|---|---|
| | Serial | Batched | Serial | Batched |
| $p_\sigma^{\text{mc}}(n=10000)$ | 1:41:11 | 1:14:38 | 0:19:56 | 0:00:35 |
| $p_\sigma^{\text{taylor}}$ | 0:00:08 | 0:00:07 | 0:00:02 | < 0:00:01 |
| $p_\sigma^{\text{mmse}}(n=5)$ | 0:00:41 | 0:00:31 | 0:00:06 | 0:00:02 |

*Table 1.* Runtimes of $p_\sigma^{\text{robust}}$ estimators (H:M:S). Each estimator computes $p_{\sigma=0.1}^{\text{robust}}$ for the CIFAR10 ResNet18 model for 50 points using the minimum number of samples $n$ necessary for convergence. The analytical estimators ($p_\sigma^{\text{taylor}}$ and $p_\sigma^{\text{mmse}}$) are more efficient than the naïve estimator ($p_\sigma^{\text{mc}}$).

$p_\sigma^{\text{mc}}$ at a given $n$ and $p_\sigma^{\text{mc}}$ at $n = 50,000$. Results for the CIFAR10 ResNet18 model are shown in Figure 4. The results indicate that $p_\sigma^{\text{mc}}$ requires around 10,000 samples per point to converge, which is impractical.

**The analytical estimators are more efficient than the naïve estimator.** Next, we examine the efficiency of the estimators by measuring their runtimes when calculating $p_{\sigma=0.1}^{\text{robust}}$ for the CIFAR10 ResNet18 model for 50 points. Runtimes are displayed in Table 1. They indicate that $p_\sigma^{\text{taylor}}$ and $p_\sigma^{\text{mmse}}$ perform 35x and 17x faster than $p_\sigma^{\text{mc}}$, respectively. Additional runtimes are in Appendix A.4.

### 4.3. Comparison of local robustness and softmax probability

**Local robustness and softmax probability are two distinct measures.** To examine the relationship between $p_\sigma^{\text{robust}}$ and $p_T^{\text{softmax}}$, we calculate $p_\sigma^{\text{mmse}}$ and $p_T^{\text{softmax}}$ for CIFAR10 and CIFAR100 models of varying levels of robustness, and measure the correlation of their values and ranks using Pearson and Spearman correlations. For a non-robust model, $p_\sigma^{\text{robust}}$ and $p_T^{\text{softmax}}$ are not strongly correlated (Figure 5). As model robustness increases, the two quantities become more correlated (Figures 6 and 7). However, even for robust models, the relationship between the two quantities is mild (Figure 7). That $p_\sigma^{\text{robust}}$ and $p_T^{\text{softmax}}$ are not

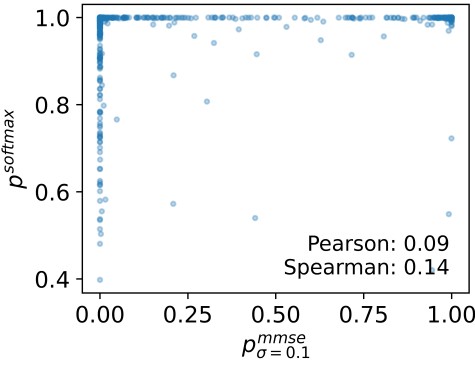

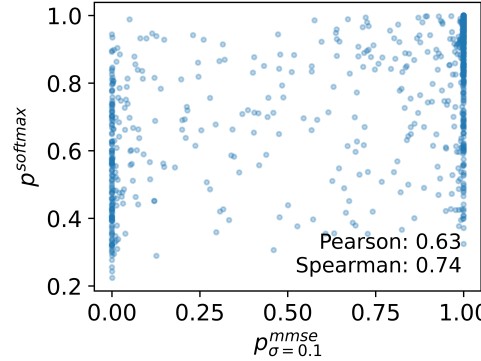

*Figure 5.* Relationship between $p_\sigma^{\mathrm{robust}}$ and $p_T^{\mathrm{softmax}}$ for a non-robust CIFAR10 ResNet18 model ($\lambda = 0$). For a non-robust model, $p_\sigma^{\mathrm{robust}}$ and $p_T^{\mathrm{softmax}}$ are not strongly correlated.

*Figure 7.* Relationship between $p_\sigma^{\mathrm{robust}}$ and $p_T^{\mathrm{softmax}}$ for a robust CIFAR10 ResNet18 model ($\lambda = 0.01$). Although $p_\sigma^{\mathrm{robust}}$ and $p_T^{\mathrm{softmax}}$ become more correlated as model robustness increases (Figure 6), even for robust models, the relationship between $p_\sigma^{\mathrm{robust}}$ and $p_T^{\mathrm{softmax}}$ is mild. These results indicate that, consistent with the theory in Section 3, $p_T^{\mathrm{softmax}}$ is not a good estimator for $p_\sigma^{\mathrm{robust}}$ in general settings.

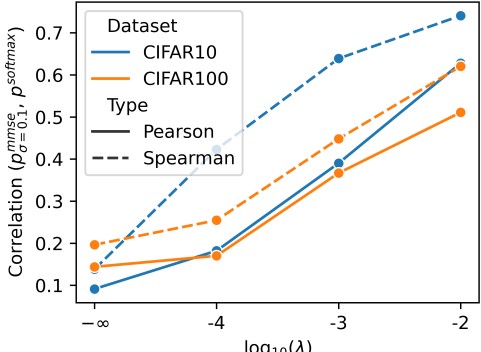

*Figure 6.* Relationship between $p_\sigma^{\mathrm{robust}}$ and $p_T^{\mathrm{softmax}}$ for the CIFAR10 ResNet18 and CIFAR100 ResNet18 models. As model robustness increases, $p_\sigma^{\mathrm{robust}}$ and $p_T^{\mathrm{softmax}}$ become more correlated.

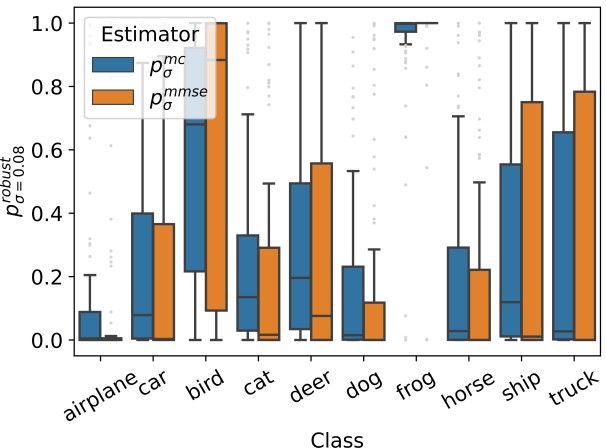

*Figure 8.* Local robustness bias among classes (CIFAR10 ResNet18). $p_\sigma^{\mathrm{robust}}$ reveals that the model is less locally robust for some classes than for others. The analytical estimator $p_\sigma^{\mathrm{mmse}}$ properly captures this model bias.

strongly correlated is consistent with the theory in Section 3: in general settings, $p_T^{\mathrm{softmax}}$ is not a good estimator for $p_\sigma^{\mathrm{robust}}$.

### 4.4. Applications of local robustness

**$p_\sigma^{\mathrm{robust}}$ detects local robustness bias.** We demonstrate that $p_\sigma^{\mathrm{robust}}$ can detect bias in local robustness by calculating $p_\sigma^{\mathrm{robust}}$ using $p_\sigma^{\mathrm{mmse}}$ and examining its distribution for each class for each model and test set over different $\sigma$'s. Results for the CIFAR10 ResNet18 model are shown in Figure 8. The results indicate that classes have different $p_\sigma^{\mathrm{robust}}$ distributions, i.e., the model is more locally robust for some classes (e.g., frog) than for others (e.g., airplane). Thus, $p_\sigma^{\mathrm{robust}}$ can be applied to detect local robustness bias, which is critical when models are deployed in high-stakes, real-world settings.

**$p_\sigma^{\mathrm{robust}}$ identifies images that are robust to and images that are vulnerable to random noise.** We demonstrate that $p_\sigma^{\mathrm{robust}}$ can also distinguish between images that are robust to and images that are vulnerable to random noise by visualizing images with the highest and lowest $p_\sigma^{\mathrm{mmse}}$ in each class for each model. For comparison, we do the same with $p_T^{\mathrm{softmax}}$. Example CIFAR10 images are displayed in Figure 9. Images with low $p_\sigma^{\mathrm{robust}}$ tend to have neutral colors, with the object being a similar color as the background (making the prediction likely to change when the image is slightly perturbed), while images with high $p_\sigma^{\mathrm{robust}}$ tend to be brightly-colored, with the object strongly contrasting

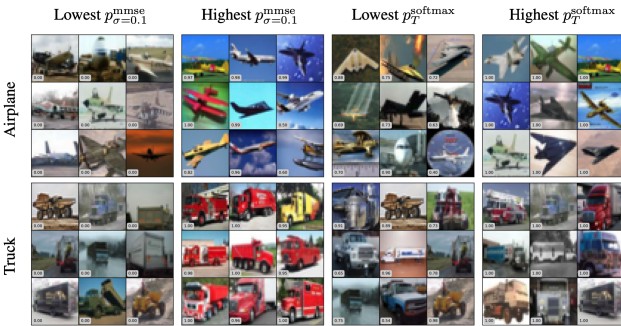

*Figure 9.* Images with the lowest and highest $p_\sigma^{\mathrm{robust}}$ among CI-FAR10 classes. Images with high $p_\sigma^{\mathrm{robust}}$ are brighter with stronger object-background contrast (making them more robust to random noise) than those with low $p_\sigma^{\mathrm{robust}}$. This difference is less evident for $p_T^{\mathrm{softmax}}$.

with the background (making the prediction likely to stay constant when the image is slightly perturbed). These differences are not as evident for images with the highest and lowest $p_T^{\mathrm{softmax}}$. Thus, in addition to detecting local robustness bias, $p_\sigma^{\mathrm{robust}}$ can also be applied to identify images that are robust to and images that are vulnerable to random noise.

For all experiments described above, full results are in Appendix A.4.

## 5. Conclusion

In this work, we take the first steps towards estimating local model robustness. We show that the naïve approach is inefficient. To address this problem, we develop efficient analytical estimators using local linear function approximation and the multivariate Normal CDF, and empirically confirm their accuracy and efficiency. Then, we demonstrate the usefulness of these estimators in performing various tasks such as measuring robustness bias and identifying examples vulnerable to noise.

To our knowledge, this work is the first to investigate local robustness in a multi-class setting and develop efficient analytical estimators for local robustness. The analytical aspect of these estimators not only advances conceptual understanding of local robustness, connecting it to randomized smoothing and softmax probability, but also enables local robustness to be used in applications that require differentiability. In addition, the efficiency of these estimators makes the computation of local robustness practical, enabling tasks that help with model debugging and establishing user trust.

One limitation of this work is that it focuses on local robustness for classification. The estimators may also provide inexact estimates that affect downstream decisions (as is typical of any inexact estimator). Defining local robustness for regression and developing efficient analytical estimators

for this setting represent future research directions. Other future research directions include exploring additional applications of local robustness, such as using local robustness to perform uncertainty calibration and to train locally robust models.

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

# A. Appendix

## A.1. Proofs

**Lemma A.1.** *The local robustness of a multi-class linear model $f(\mathbf{x}) = \mathbf{w}^\top \mathbf{x} + b$, with $\mathbf{w} \in \mathbb{R}^{d \times C}$ and $b \in \mathbb{R}^C$, with respect to a target class $t$ is given by the following. Define the decision boundary weights $w_i' = w_t - w_i \in \mathbb{R}^d, \forall i \neq t$, where $w_t, w_i$ are rows of $\mathbf{w}$ and biases $b_i' = (w_t' - w_i')^\top \mathbf{x} + (b_t - b_i) \in \mathbb{R}$, then*

$$p_\sigma^{robust}(\mathbf{x}) = CDF_{\mathcal{N}(0, UU^\top)}\left( \frac{b_1'}{\sigma \|w_1'\|_2}, \dots \frac{b_i'}{\sigma \|w_i'\|_2}, \dots \frac{b_C'}{\sigma \|w_C'\|_2} \right)$$

$$where \quad U = \left[ \frac{w_1'}{\|w_1'\|_2}; \dots \frac{w_i'}{\|w_i'\|_2}; \dots \frac{w_C'}{\|w_C'\|_2} \right] \in \mathbb{R}^{(C-1) \times d}$$

*and $CDF_{\mathcal{N}(0, UU^\top)}$ refers to the $(C-1)$-dimensional Normal CDF with covariance $UU^\top$.*

*Proof.* First, we rewrite $p_\sigma^{\text{robust}}$ in the following manner, by defining $g_i(\mathbf{x}) = f_t(\mathbf{x}) - f_i(\mathbf{x}) > 0$, which is the "decision boundary function".

$$p_\sigma^{\text{robust}} = P_{\epsilon \sim \mathcal{N}(0, \sigma^2)}\left[ \max_i f_i(\mathbf{x} + \epsilon) < f_t(\mathbf{x} + \epsilon) \right] = P_{\epsilon \sim \mathcal{N}(0, \sigma^2)}\left[ \bigcup_{i=1; i \neq t}^{C} g_i(\mathbf{x} + \epsilon) > 0 \right]$$

Now, assuming that $f, g$ are linear such that $g_i(\mathbf{x}) = w_i'^\top \mathbf{x} + g(0)$, we have $g_i(\mathbf{x} + \epsilon) = g_i(\mathbf{x}) + w_i'^\top \epsilon$, and obtain

$$p_\sigma^{\text{robust}} = P_{\epsilon \sim \mathcal{N}(0, \sigma^2)}\left[ \bigcup_{i=1; i \neq t}^{C} w_i'^\top \epsilon > -g_i(\mathbf{x}) \right]$$

$$= P_{z \sim \mathcal{N}(0, I_d)}\left[ \bigcup_{i=1; i \neq t}^{C} \frac{w_i'^\top}{\|w_i'\|_2} z > -\frac{g_i(\mathbf{x})}{\sigma \|w_i'\|_2} \right] \qquad \text{(Rescaling and standardization)}$$

We now make the following observations:

- For any matrix $U \in \mathbb{R}^{C \times d}$ and a d-dimensional Gaussian random variable $z \sim \mathcal{N}(0, I_d) \in \mathbb{R}^d$, we have $U^\top z \sim \mathcal{N}(0, UU^\top)$, i.e., an C-dimensional Gaussian random variable.

- CDF of a multivariate Gaussian RV is defined as $P_z[\bigcup_i z_i < t_i]$ for some input values $t_i$

Using these observations, if we construct $U = [\frac{w_1'}{\|w_1'\|_2}; \frac{w_2'}{\|w_2'\|_2}; \dots \frac{w_C'}{\|w_C'\|_2}] \in \mathbb{R}^{(C-1) \times d}$, and obtain

$$p_{\text{robust}} = P_{u \sim \mathcal{N}(0, UU^\top)}\left[ \bigcup_{i=1; i \neq t}^{C} u_i < \frac{g_i(\mathbf{x})}{\sigma \|w_i'\|_2} \right]$$

$$= CDF_{\mathcal{N}(0, UU^\top)}\left( \left[ \frac{g_1(\mathbf{x})}{\sigma \|w_1'\|_2}, \frac{g_2(\mathbf{x})}{\sigma \|w_2'\|_2}, \dots \frac{g_C(\mathbf{x})}{\sigma \|w_C'\|_2} \right] \right)$$

where $g_i(\mathbf{x}) = w_i'^\top \mathbf{x} + g_i(0) = (w_t' - w_i')^\top \mathbf{x} + (b_t - b_i)$

$\square$

*Proposition* 3. The Taylor estimator for a model $f$ and point $\mathbf{x}$ is given by linearizing $f$ around $\mathbf{x}$ using $\mathbf{w} = \nabla_{\mathbf{x}} f(\mathbf{x})$ and $b = f(\mathbf{x})$, with decision boundaries $g_i(\mathbf{x}) = f_t(\mathbf{x}) - f_i(\mathbf{x})$, leading to

$$p_\sigma^{\text{taylor}}(\mathbf{x}) = \text{CDF}_{\mathcal{N}(0, UU^\top)}\left(\left[\frac{g_1(\mathbf{x})}{\sigma\|\nabla_{\mathbf{x}} g_1(\mathbf{x})\|_2}, \dots \frac{g_i(\mathbf{x})}{\sigma\|\nabla_{\mathbf{x}} g_i(\mathbf{x})\|_2}, \dots \frac{g_C(\mathbf{x})}{\sigma\|\nabla_{\mathbf{x}} g_C(\mathbf{x})\|_2}\right]\right)$$

with $U \in \mathbb{R}^{(C-1) \times d}$ defined as in the linear case.

*Proof.* Using the notations from the previous Lemma 3.1, we can use $g(\mathbf{x} + \epsilon) \approx g(\mathbf{x}) + \nabla_{\mathbf{x}} g(\mathbf{x})^\top \epsilon$ using a first order Taylor series expansion. Thus we use $w_i' = \nabla_{\mathbf{x}} g_i(\mathbf{x})$ and $b' = g(\mathbf{x})$, and plug it into the result of Lemma 3.1. □

*Proposition* 4. The MMSE estimator for a model $f$ and point $\mathbf{x}$ is given by linearizing $f$ around $\mathbf{x}$ using $\mathbf{w} = \sum_{j=1}^N \nabla_{\mathbf{x}} f(\mathbf{x} + \epsilon)$ and $b = \sum_{j=1}^N f(\mathbf{x})$, with decision boundaries $g_i(\mathbf{x}) = f_t(\mathbf{x}) - f_i(\mathbf{x})$, leading to

$$p_\sigma^{\text{mmse}}(\mathbf{x}) = \text{CDF}_{\mathcal{N}(0, UU^\top)}\left(\left[\frac{\sum_{j=1}^N g_1(\mathbf{x} + \epsilon)}{\sigma\|\sum_{j=1}^N \nabla_{\mathbf{x}} g_1(\mathbf{x} + \epsilon)\|_2}, \dots \frac{\sum_{j=1}^N g_C(\mathbf{x} + \epsilon)}{\sigma\|\sum_{j=1}^N \nabla_{\mathbf{x}} g_C(\mathbf{x} + \epsilon)\|_2}\right]\right)$$

with $U \in \mathbb{R}^{(C-1) \times d}$ defined as in the linear case, where $N$ is the number of perturbations.

*Proof.* We would like to improve upon the Taylor approximation to $g(\mathbf{x} + \epsilon)$ by using an MMSE local function approximation. Essentially, we'd like the find $w \in \mathbb{R}^d$ and $b \in \mathbb{R}$ such that

$$(w^*(\mathbf{x}), b^*(\mathbf{x})) = \arg\min_{w,b} \mathbb{E}_{\epsilon \sim \mathcal{N}(0, \sigma^2)} (g(x + \epsilon) - w^\top \epsilon - b)^2$$

A straightforward solution by finding critical points and equating it to zero gives us the following:

$$w^*(\mathbf{x}) = \mathbb{E}_\epsilon\left[g(x + \epsilon)\epsilon^\top\right]/\sigma^2 = \mathbb{E}_\epsilon\left[\nabla_{\mathbf{x}} g(\mathbf{x} + \epsilon)\right] \quad \text{(Stein's Lemma)}$$

$$b^*(\mathbf{x}) = \mathbb{E}_\epsilon g(x + \epsilon)$$

Plugging in these values of $w^*, b^*$ into Lemma 3.1, we have the result.

□

## A.2. Datasets

The MNIST dataset consists of images of gray-scale handwritten digits. The images span 10 classes: digits 0 through 9. Each image is of size 28 pixels x 28 pixels. The training set consists of 60,000 images and the test set consists of 10,000 images.

The FashionMNIST dataset consists of gray-scale images of articles of clothing. The images span 10 classes: t-shirt, trousers, pullover, dress, coat, sandal, shirt, sneaker, bag, and ankle boot. Each image is of size 28 pixels x 28 pixels. The training set consists of 60,000 images and the test set consists of 10,000 images.

The CIFAR10 dataset consists of color images of common objects and animals. The images span 10 classes: airplane, car, bird, cat, deer, dog, frog, horse, ship, and truck. Each image is of size 3 pixels x 32 pixels x 32 pixels. The training set consists of 50,000 images and the test set consists of 10,000 images.

The CIFAR100 dataset consists of color images of common objects and animals. The images span 100 classes: apple, bowl, chair, dolphin, lamp, mouse, plain, rose, squirrel, train, etc. Each image is of size 3 pixels x 32 pixels x 32 pixels. The training set consists of 50,000 images and the test set consists of 10,000 images.

For experiments, we use 1,000 randomly-selected test set images for each dataset.

### A.3. Models

For the MNIST and FashionMNIST (FMNIST) datasets, we train a linear model and a convolutional neural network (CNN) to perform 10-class classification. The linear model consists of one hidden layer with 10 neurons. The CNN consists of four hidden layers: one convolutional layer with 5x5 filters and 10 output channels, one convolutional layer 5x5 filters and 20 output channels, and one linear layer with 50 neurons, and one linear layer 10 neurons.

For CIFAR10 and CIFAR100 datasets, we train a ResNet18 model to perform 10-class and 100-class classification, respectively. The model architecture is described in (He et al., 2016). We train the ResNet18 models using varying levels of gradient norm regularization to obtain models with varying levels of robustness. The larger the weight of gradient norm regularization ($\lambda$), the more robust the model.

All models were trained using stochastic gradient descent. Hyperparameters were selected to achieve decent model performance. The emphasis is on analyzing the estimators' estimates of local robustness of each model, not on high model performance. Thus, we do not focus on tuning model hyperparameters. All models were trained for 200 epochs. The test set accuracy (on each dataset's full 10,000-point test set) for each model is shown in Table 2.

| Dataset | Model | $\lambda$ | Test set accuracy |
|---|---|---|---|
| MNIST | Linear | 0 | 92% |
| MNIST | CNN | 0 | 99% |
| FashionMNIST | Linear | 0 | 84% |
| FashionMNIST | CNN | 0 | 91% |
| CIFAR10 | ResNet18 | 0 | 94% |
| CIFAR10 | ResNet18 | 0.0001 | 93% |
| CIFAR10 | ResNet18 | 0.001 | 90% |
| CIFAR10 | ResNet18 | 0.01 | 85% |
| CIFAR100 | ResNet18 | 0 | 76% |
| CIFAR100 | ResNet18 | 0.0001 | 74% |
| CIFAR100 | ResNet18 | 0.001 | 69% |
| CIFAR100 | ResNet18 | 0.01 | 60% |

*Table 2.* Accuracy of models on test set.

## A.4. Experiments

### A.4.1. CONVERGENCE OF $p_\sigma^{\mathrm{mc}}$

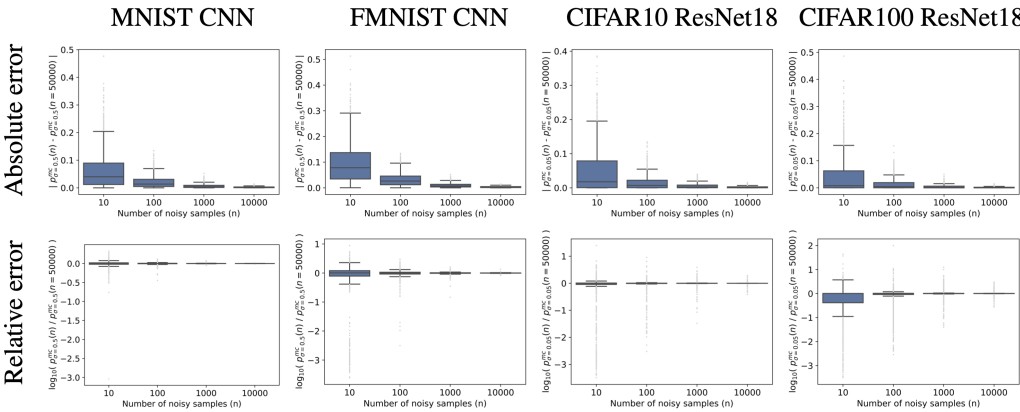

*Figure 10.* Convergence of $p_\sigma^{\mathrm{mc}}$.

### A.4.2. CONVERGENCE OF $p_\sigma^{\mathrm{mmse}}$

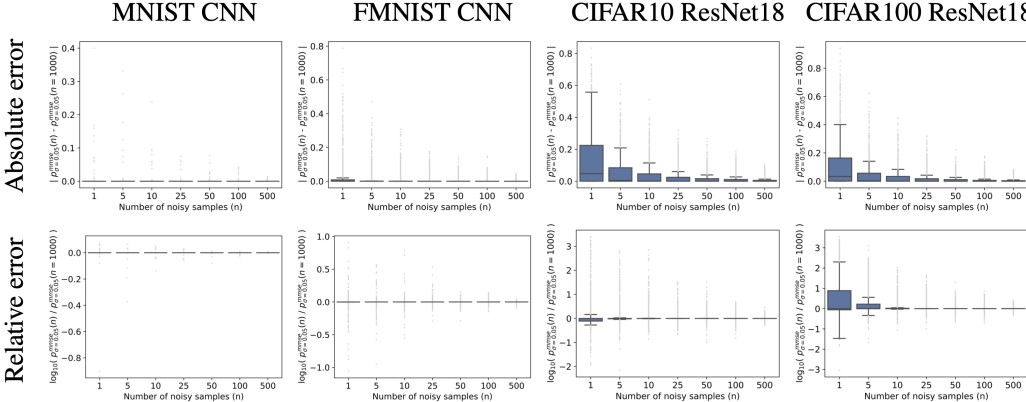

*Figure 11.* Convergence of $p_\sigma^{\mathrm{mmse}}$.

### A.4.3. DISTRIBUTION OF $p_\sigma^{\text{robust}}$ OVER NOISE

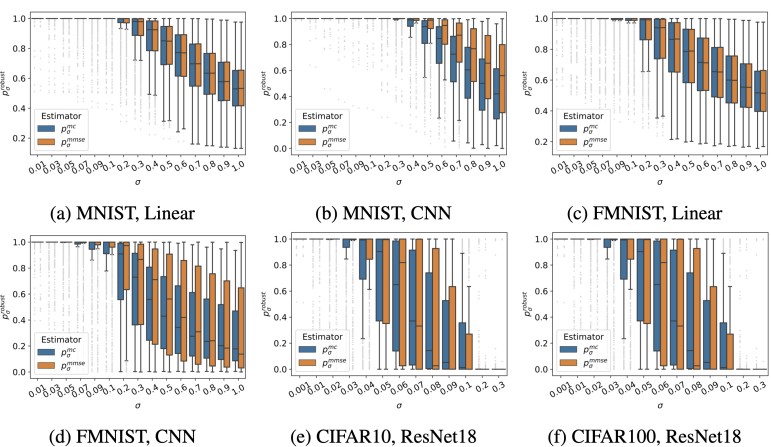

(a) MNIST, Linear      (b) MNIST, CNN      (c) FMNIST, Linear

(d) FMNIST, CNN      (e) CIFAR10, ResNet18      (f) CIFAR100, ResNet18

*Figure 12.* Distribution of $p_\sigma^{\text{robust}}$ over $\sigma$.

### A.4.4. ACCURACY OF $p_\sigma^{\text{robust}}$ ESTIMATORS

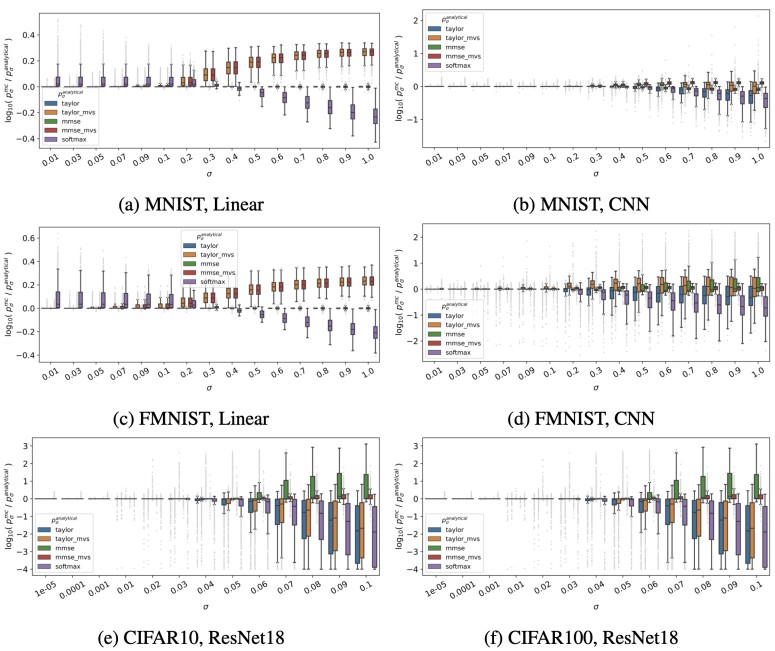

(a) MNIST, Linear      (b) MNIST, CNN

(c) FMNIST, Linear      (d) FMNIST, CNN

(e) CIFAR10, ResNet18      (f) CIFAR100, ResNet18

*Figure 13.* Accuracy of $p_\sigma^{\text{robust}}$ estimators over $\sigma$.

### A.4.5. ACCURACY OF $p_\sigma^{\text{robust}}$ ESTIMATORS FOR ROBUST MODELS

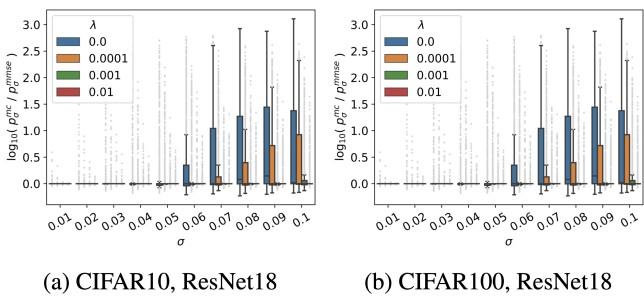

(a) CIFAR10, ResNet18          (b) CIFAR100, ResNet18

*Figure 14.* Accuracy of $p_\sigma^{\text{robust}}$ estimators over $\sigma$ for robust models.

### A.4.6. MV-SIGMOID FUNCTION'S APPROXIMATION OF MVN-CDF FUNCTION

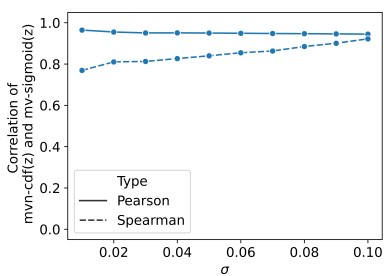

*Figure 15.* mv-sigmoid function's approximation of mvn-cdf function over $\sigma$.

### A.4.7. LOCAL ROBUSTNESS BIAS AMONG CLASSES

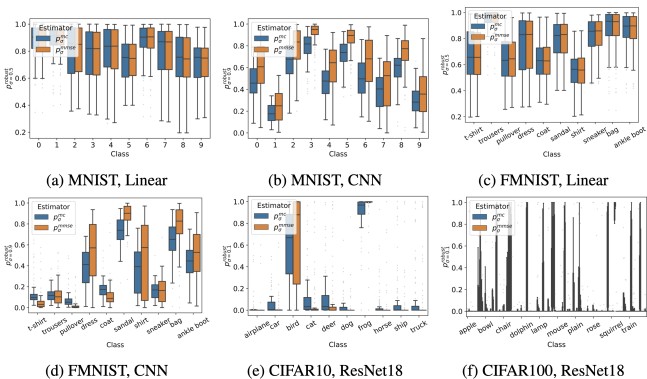

(a) MNIST, Linear          (b) MNIST, CNN          (c) FMNIST, Linear

(d) FMNIST, CNN          (e) CIFAR10, ResNet18          (f) CIFAR100, ResNet18

*Figure 16.* Distribution of $p_\sigma^{\text{robust}}$ for across classes.

A.4.8. RUNTIMES OF $p_\sigma^{\text{robust}}$ ESTIMATORS

| Estimator | # samples ($n$) | CPU: Intel x86_64 | | GPU: Tesla V100-PCIE-32GB | |
|---|---|---|---|---|---|
| | | Serial | Batched | Serial | Batched |
| $p_\sigma^{\text{mc}}$ | $n = 100$ | 0:00:59 | 0:00:42 | 0:00:12 | 0:00:01 |
| | $n = 1000$ | 0:09:50 | 0:07:22 | 0:02:00 | 0:00:04 |
| | $n = 10000$ | *1:41:11* | *1:14:38* | *0:19:56* | *0:00:35* |
| $p_\sigma^{\text{taylor}}$ | N/A | 0:00:08 | 0:00:07 | 0:00:02 | < 0:00:01 |
| $p_\sigma^{\text{taylor\_mvs}}$ | N/A | 0:00:08 | 0:00:07 | 0:00:01 | < 0:00:01 |
| $p_\sigma^{\text{mmse}}$ | $n = 1$ | 0:00:08 | 0:00:10 | 0:00:02 | 0:00:02 |
| | $n = 5$ | *0:00:41* | *0:00:31* | *0:00:06* | *0:00:02* |
| | $n = 10$ | 0:01:21 | 0:01:02 | 0:00:11 | 0:00:02 |
| | $n = 25$ | 0:03:21 | 0:02:44 | 0:00:26 | 0:00:03 |
| | $n = 50$ | 0:06:47 | 0:05:38 | 0:00:51 | 0:00:04 |
| | $n = 100$ | 0:13:57 | 0:11:31 | 0:01:42 | 0:00:06 |
| $p_\sigma^{\text{mmse\_mvs}}$ | $n = 1$ | 0:00:08 | 0:00:08 | 0:00:01 | 0:00:01 |
| | $n = 5$ | *0:00:41* | *0:00:32* | *0:00:05* | *0:00:01* |
| | $n = 10$ | 0:01:21 | 0:01:00 | 0:00:10 | 0:00:02 |
| | $n = 25$ | 0:03:24 | 0:02:37 | 0:00:25 | 0:00:02 |
| | $n = 50$ | 0:06:47 | 0:05:35 | 0:00:51 | 0:00:03 |
| | $n = 100$ | 0:13:28 | 0:11:32 | 0:01:42 | 0:00:06 |
| $p_T^{\text{softmax}}$ | N/A | 0:00:01 | < 0:00:01 | < 0:00:01 | < 0:00:01 |

*Table 3.* Runtimes of each $p_\sigma^{\text{robust}}$ estimator. Each estimator computes $p_{\sigma=0.1}^{\text{robust}}$ for the CIFAR10 ResNet18 model for 50 data points. For estimators that use sampling, the row with the minimum number of samples necessary for convergence is italicized. The analytical estimators ($p_\sigma^{\text{taylor}}$, $p_\sigma^{\text{taylor\_mvs}}$, $p_\sigma^{\text{mmse}}$, and $p_\sigma^{\text{mmse\_mvs}}$) are more efficient than the naïve estimator ($p_\sigma^{\text{mc}}$). Runtimes are in the format of hour:minute:second.

## A.5. Broader Impact

This work is concerned with improving estimation of local robustness of machine learning models, and as such does not have any immediate foreseeable negative societal impact. However, inexact estimation can affect downstream decisions, and as such, estimator quality must always be taken into account to mitigate such cases.

