# OpenReview forum: "Efficient Estimation of Local Robustness of Machine Learning Models"
_ICML.cc/2023/Workshop/IMLH — IMLH 2023 Poster_

### Official Review · Reviewer_pDrf · 2023-06-10
**The work is significant in advancing the understanding and practical applications of local robustness in machine learning models. By introducing efficient analytical estimators, it overcomes the limitations of the na¨ıve Monte-Carlo sampling approach, enabling practical computation of local robustness. The applications of the estimators in detecting robustness bias and identifying vulnerable examples highlight their utility in real-world scenarios, such as model debugging and establishing user trust. Overall, the work contributes to the field of machine learning interpretability and model reliability.**

**Rating:** 6
**Confidence:** 3

**Review:**


Pros:

Introduction of novel analytical estimators for local robustness.
Empirical validation and evaluation of the estimators.
Exploration of connections between local robustness and softmax probability.
Applications of the estimators in detecting robustness bias and identifying vulnerable examples.
Clear problem statement and well-documented experiments and results.


Cons:


The paper is generally clear in its presentation. It provides a clear problem statement and defines the concepts related to local robustness. The derivations of the analytical estimators are explained step by step, and the empirical evaluations are well-described. However, some parts could benefit from additional explanations, especially in the mathematical derivations, to improve understanding for readers who may not be familiar with the specific techniques and terminology used.

---

### Meta-Review · Area_Chair_o2Fu · 2023-06-19

**Recommendation:** Accept (Poster)
**Confidence:** 4

**Metareview:**

The paper studies practical implementation of local robustness in machine learning models. It addresses the limitations of the conventional Monte-Carlo sampling method by introducing efficient analytical estimators, allowing for practical computation of local robustness. The application of these estimators in detecting robustness bias and identifying vulnerable examples demonstrates their usefulness in real-world situations, such as model debugging and building user trust.

Overall, this work enhances our understanding of machine learning interpretability and improves model reliability. The reviewers pointed out a few weaknesses, such as the necessity of further explanation, but agreed that this is a great contribution to the healthcare and interpretability field. Given these, the AC recommend acceptance.

---

### Decision · Program_Chairs · 2023-06-20

Accept (Poster)